# Association of Human Leukocyte Antigens Class II Variants with Susceptibility to Hidradenitis Suppurativa in a Caucasian Spanish Population

**DOI:** 10.3390/jcm9103095

**Published:** 2020-09-25

**Authors:** J. Gonzalo Ocejo-Vinyals, Miguel A. Gonzalez-Gay, Marcelo A. Fernández-Viña, Juan Cantos-Mansilla, Iosune Vilanova, Ricardo Blanco, Marcos A. González-López

**Affiliations:** 1Department of Immunology, Hospital Universitario Marqués de Valdecilla. Avda. de Valdecilla s/n, 39008 Santander, Cantabria, Spain; juan.cantos1986@gmail.com; 2Department of Rheumatology, Hospital Universitario Marqués de Valdecilla, Universidad de Cantabria, IDIVAL, 39008 Santander, Cantabria, Spain; miguelaggay@hotmail.com (M.A.G.-G.); ricardo.blanco@scsalud.es (R.B.); 3Histocompatibility, Immunogenetics & Disease Profiling Laboratory, Stanford Blood Center, Palo Alto, CA 94304, USA; marcelof@stanford.edu; 4Department of Dermatology, Hospital Universitario Marqués de Valdecilla, Universidad de Cantabria, IDIVAL, 39008 Santander, Cantabria, Spain; iosunevilanova@gmail.com (I.V.); e204@scsalud.es (M.A.G.-L)

**Keywords:** hidradenitis suppurativa, HLA, allele, haplotype

## Abstract

Hidradenitis suppurativa (HS) is a chronic inflammatory cutaneous disease of the hair follicle typically presenting recurrent, painful, and inflamed lesions on the inverse areas of the body. Although its pathogenesis remains unknown, the immune system appears to play a potential role. To date, two previous studies have not found any association between the Human Leukocyte Antigen system (HLA) and HS. In this study we analyzed the *HLA-A*, *-B, -C*; and *DRB1*, -*DQA1,* and *–DQB1* allele distribution in 106 HS patients and 262 healthy controls from a Caucasian population in Cantabria (northern Spain). *HLA-A**29 and *B**50 were significantly more common in HS patients and *A**30 and *B**37 in controls, but these associations disappeared after statistical correction. *DRB1**07, *DQA1**02, and *DQB1**02 were significantly more common in controls (*p* 0.026, *p* 0.0012, and *p* 0.0005, respectively) and the HLA allele *DQB1**03:01 was significantly more common in HS patients (*p* 0.00007) after the Bonferroni correction. The *DRB1*07~DQA1*02~DQB1*02* haplotype was significantly more common in controls (*p* < 0.0005). This is the first study showing an association between HLA-class II and HS. Our results suggest that HLA-II alleles (*DRB1**07, *DQA1**02, *DQB1**02, and *DQB1**03:01) and the *DRB1**07~*DQA1**02~*DQB1**02 haplotype could influence resistance or susceptibility to HS.

## 1. Introduction

Hidradenitis suppurativa (HS), or acne inversa, is a debilitating and recurrent chronic inflammatory cutaneous disease that usually presents after puberty with painful, inflamed lesions on the apocrine gland-bearing areas of the body, most commonly the axillae, inguinal, and anogenital regions [1]. HS affects mostly young adults and the female-to-male ratio is approximately 3:1 [2]. The estimated prevalence of HS in the Spanish population is 1% (about half a million people). Although its etiopathogenesis remains uncertain, follicular occlusion due to hyperkeratosis is thought to be the primary event. Several factors such as cigarette smoking, obesity, diabetes mellitus, ethnicity, stress, and drugs have been related to an increased predisposition to HS [2,3,4,5,6,7,8,9].

HS has been reported in several members of different generations of various families [10,11]. Moreover, loss-of-function mutations in the genes encoding essential components of the γ-secretase multiprotein complex (*PSEN1*, *PSENEN,* and *NCSTN* genes) consistent with an autosomal dominant inheritance model have been reported in affected families [12,13,14,15]. The role of bacteria in the pathogenesis of HS is controversial. It has been suggested that these infectious agents may play a role in disease progression rather that in the pathogenesis itself [2,16].

The human leukocyte antigen (HLA) system plays a critical role in controlling the immune response and has been involved in the pathogenesis of many diseases, in particular those involving autoimmune phenomena [17]. HLA-encoded factors were found to be associated with several dermatologic diseases, including psoriasis, autoimmune blistering diseases, alopecia areata, dermatitis herpetiformis, drug-induced cutaneous hypersensitivity, and others [18,19,20,21,22].

To date, only two studies have investigated the potential role of class I and class II HLA in the pathogenesis of HS, with contradictory findings [23,24]. In 1998, O’Loughlin et al. [23] reported the distribution of HLA-A and -B antigens in 27 Irish patients with HS using serological methods. These investigators found an increased frequency of HLA-A1 and HLA-B8 in patients with moderate and severe clinical disease. Shortly after, Lapins et al. [24] analyzed the frequencies of *HLA-A*, *-B,* and *HLA-DR* alleles in 42 unrelated Swedish patients with HS and 250 healthy controls. These results found no association between any HLA alleles and genetic susceptibility to HS. The discrepancies between the studies may have resulted from limited sample sizes. Many years have passed since any group has studied the relationship between HLA and HS.

To further investigate the potential implication of the HLA region in susceptibility to HS, we assessed the distribution of alleles at *HLA-A*, *B*, *C*, *DRB1*, *DQA1*, and *DQB1* loci in HS patients and healthy controls from a genetically homogeneous population (Cantabria, northern Spain) [25,26]. The results obtained after analysis of the data of the HLA alleles and haplotypes were used to conduct a comparison between individuals with increased or reduced susceptibility to HS.

## 2. Experimental Section

### 2.1. Subjects

A total of 106 HS patients and 262 healthy blood donors were recruited through a retrospective case-control study. HS was diagnosed by dermatologists, and all patients fulfilled the established diagnostic criteria for HS [27]. Neither consanguinity nor relationship existed among patients or healthy controls. Furthermore, patients with a family history of HS, diabetes mellitus, or other autoimmune diseases were excluded. The blood donors (mean age, 48 years; range, 18–65 years; male/female ratio, 1:3) and HS patients (mean age, 56 years; range, 23–76 years) were of Caucasian background, all of them belonging to the Community of Cantabria (northern Spain). No other ethnicities were included in the patient population. Healthy control blood donors were examined by a dermatologist to rule out the presence of skin lesions that could bias the study. Procedures used in the study conformed to the principles outlined in the Declaration of Helsinki. All samples were collected with the written consent of the participants. The protocol of the study was accepted and approved by the Research Ethics Board of Hospital Universitario Marqués de Valdecilla.

### 2.2. HLA Typing

High-molecular-weight DNA samples were prepared from whole blood by using the Maxwell 16 Blood DNA Purification Kit (Promega Biotech Ibérica, S.L., Madrid, Spain). DNA-based HLA classes I and II typing was performed using sequence-specific oligoprobes in the Luminex 100 system (Luminex, Austin, TX, USA) and the Lifecodes HLA Typing Kits, and analyzed using the MatchIT software (Gen-Probe Inc., San Diego, CA, USA) following the manufacturer’s instructions. To confirm the results, HLA-DQB1 high-resolution typing was performed through sequencing-based typing (SBT) using the SBTexcellerator Kit and analyzed with the SBTengine-SBT HLA typing software (GenDx, Utrecht, The Netherlands).

### 2.3. Statistical Analysis

HLA allelic frequencies were calculated by direct counting. Haplotype frequencies were analyzed by the maximum-likelihood method assuming Hardy–Weinberg equilibrium (HWE). Patients and controls were compared using the two-tailed χ^2^ test, or the Fisher’s exact test when necessary with Yates’ continuity correction with an α level of 0.05. The *p*-values and odds ratios (OR) with 95% confidence intervals (CIs) were calculated using SPSS version 12 (SPSS Inc., Chicago, IL, USA). A *p*-value < 0.05 was considered statistically significant. The Bonferroni correction for multiple comparisons was applied to avoid false positive results. It was calculated by multiplying the initial *p*-value by the number of HLA alleles at each locus. A value of *p* < 0.05 was considered statistically significant. The statistical power of the study’s sample size was calculated using PS Power and Sample Size Calculations software version 3.0 (NCSS Statistical Analysis & Graphics Software, Kaysville, UT, USA) [28].

## 3. Results

### 3.1. HLA-A, -B, and -C Alleles in HS Patients and Controls

All allele and haplotype frequencies were in HWE. The frequencies of HLA-A, -B, and -C alleles in HS patients and healthy controls are summarized in Table 1. HLA-A*29 was significantly more common in healthy controls and HLA-A*30 in HS patients (*p* 0.02, OR 0.44, 95% CI: 0.21–0.91 and *p* 0.03, OR 2.20, 95% CI: 1.13–4.28, respectively). HLA-B*37 was also significantly more common in HS patients and HLA-B*50 in healthy controls (*p* 0.04, OR 3.64, 95% CI: 1.08–12.24 and *p* 0.04, OR 0.38, 95% CI: 0.15–0.92, respectively). These associations were no longer significant after performing the Bonferroni correction.

### 3.2. HLA-DRB1, -DQA1, and -DQB1 Alleles and Haplotypes in HS Patients and Controls

The examination of the distribution of HLA class II alleles showed that the allele groups including DRB1*07, DQA1*02, and DQB1*02 alleles were significantly more common in healthy controls (*p* 0.002, OR 0.45, 95% CI: 0.27–0.74; *p* 0.0002, OR 0.43 95% CI: 0.28–0.67; and *p* 0.0001, OR 0.44, 95% CI: 0.27–0.67, respectively; Table 2). These associations remained statistically significant after the Bonferroni correction (*p* 0.026, 0.0012, and 0.0006, respectively).

In contrast, the allele group HLA-DQB1*03 was statistically more common in HS patients (*p* 0.02, OR 1.52 95% CI 1.08–2.13). This was especially true for the HLA-DQB1*03:01 allele, which was significantly more common in HS patients than healthy controls (*p* 0.00001, OR 2.33, 95% CI: 1.57–3.44), with a statistical power >90% (Table 3a). Other DQB1*03 alleles did not show significant differences. When the various DQA1 alleles in linkage disequilibrium with DQB1*03:01 in the study’s population (DQA1*03, *05, and *06) were analyzed, the HLA DQA1*03–DQB1*03:01 and the HLA DQA1*05–DQB1*03:01 haplotype blocks, assigned by linkage disequilibrium, showed statistically significant differences in HS patients and healthy controls (*p* 0.01 and 0.004, respectively; Table 3b), but with lower *p*-values than observed for individual allele DQB1*03:01. The combination of both haplotype blocks showed higher significance (*p* 0.0001, OR 2.25, 95% CI 1.52–3.34) than that observed in the individual DQA1-DQB1 blocks bearing DQB1*03:01 with a statistical power >80%.

The analysis of the different haplotype blocks showed a significant increase in the DRB1*07–DQA1*02–DQB1*02 haplotype in controls (*p* 0.0014 OR 0.39 95% CI 0.23–0.70); this association remained significant after the Bonferroni correction (*p* 0.028, Table 4).

## 4. Discussion

This study involved the largest HS patient group assessed to identify a potential association with the HLA region. The pathophysiology of HS is still not well understood [1,2]. Epidemiological studies suggest the influence of many genetic and environmental factors. In addition to family cases due to mutations in genes encoding essential components of the γ-secretase multiprotein complex (PSEN1, PSENEN, and NCSTN genes) [12,13,14,15], several other factors including psychological influences, metabolism, smoking habits, bacterial infections, and associations with other diseases (mainly inflammatory disorders) have been related to a higher incidence of HS [1,2,3,4,5,6,7,8,9,16,29]. Recently, our group reported on the role of several adipokines (adiponectin, leptin, resistin, and visfatin) in patients with HS and investigated the possible associations with insulin resistance, HS risk, and disease severity [30]. The role of the immune system in the etiopathogenesis of HS has yet to be elucidated. Several reports have focused on the role of immune dysregulation with complement abnormalities [31,32]. Recently, innate immunity, auto-inflammatory mechanisms, and distribution of certain T-cell lymphocyte subpopulations have been shown to influence the pathophysiology of HS [33,34,35].

There was limited evidence, if any, regarding the role of HLA factors in susceptibility and resistance to HS before this study. No study addressed the possible influence of the HLA region in susceptibility to and protection against HS since 2001 [24]. No association of *HLA-A*, *-B,* and -*DRB1* alleles and HS was found in 42 unrelated Swedish patients with HS compared to 250 controls. Similar results were found in a study of the implication of the HLA-A and HLA-B antigen loci in 27 Irish patients with untreated HS compared to 20 healthy blood donors without relevant differences [23].

We analyzed the phenotype distributions of alleles for the first time, encoding six HLA loci (*HLA-A*, *-B*, *-C*, *-DRB1*, *-DQA1*, and -*DQB1*) in 106 HS patients and 262 healthy blood donors from a condensed population (Cantabria, northern Spain). The results that showed weak effects of alleles at *HLA-A* and -*B* loci were no longer significant after statistical correction. In contrast, alleles at class II loci showed statistically significant differences in the distribution of allele groups *DRB1**07, *DQA1**02, and *DQB1**02, being significantly less common in HS patients. These alleles are in strong linkage disequilibrium, conforming to a common and extremely well-conserved haplotype block that was less common in the patient group and suggesting a protective role against the development of the disease. Because of the strong linkage disequilibrium, it was not possible to pinpoint which factor of this DRB1-DQ block confers protection.

Even more relevant than the evidence of protective alleles was the presence of the association between the *HLA-DQB1**03 group—the *DQB1**03:01 allele in particular—and susceptibility to HS in Caucasian patients from northern Spain. A comparison of the amino acid sequences of the common alleles of *DQB1**03 (03:01, 03:02, and 03:03) showed that specific amino acid replacements at residue 45 and possibly residue 57 may determine the differential associations between alleles in this group.

In the study population, *DQB1**03:01, as in most European populations, was found in association with several alleles of DQA1 and DRB1 loci. The highest statistically significant association of HLA alleles present in haplotype blocks bearing *DQB*1*03:01 was observed for this allele, with no association observed for *DRB1* alleles and weak associations for *DQA1* alleles. The allele *DQB1**03:01 allele may form heterodimer molecules when pairing with the alleles *DQA1**03, *05, and *06 alleles. Only *DQB1**03:01 heterodimers with *DQA*1*03 and *05 showed significant differences between HS patients and controls. Whether the *DQB1* locus alone could explain most of the risk in our cohort of HS patients, as has been suggested in other pathologies such as narcolepsy [36,37,38] or DQ heterodimerization of *DQα*03/*DQβ*03:01 and *DQα*05/*DQβ*03:01, and therefore, the risk of the *HLA-DQB1**03:01 gene, has to be considered alongside its polymorphic partners *HLA-DQA1**03 and *05 and would need additional, larger studies.

Interestingly, *DQB1**03:01 has been reported as a susceptibility allele in another dermatological disease: bullous pemphigoid, an organ-specific autoantibody-mediated autoimmune blistering skin disorder in normal patients and in type 2 diabetes mellitus patients receiving dipeptidyl peptidase-IV inhibitors [22,39,40,41,42]. The mechanism of this association is still unclear, but it has been proposed that *DQB1**03:01 could bind bullous pemphigoid 180 (BP180, BP antigen 2, collagen XVII), which may lead to cutaneous disease [42,43,44].

The limited patient population may be a potential limitation for the study. However, our study has a number of strengths. It encompassed, to the best of our knowledge, the largest series of HS patients ever evaluated for genetic susceptibility. Moreover, it had a monocentric design with the inclusion of patients homogeneously and carefully evaluated by a group of clinicians with extensive experience with the disease. However, additional studies in Caucasians and other populations should be conducted to further support the influence of HLA in conferring susceptibility to this disease.

## 5. Conclusions

As observed in other inflammatory chronic diseases such as rheumatoid arthritis [45], this study showed for the first time a relevant contribution of HLA class II alleles and haplotypes to protection against and susceptibility to HS. Our findings open new avenues for investigating the role of HLA factors in the mechanisms leading to the development of this disease that affects more than 500,000 people in the Spanish population.

## Figures and Tables

**Table 1 jcm-09-03095-t001:** HLA-A (a), -B (b), and -C (c) allele frequencies in HS (Hidradenitis suppurativa) patients and healthy controls.

(**a**)					
**HLA-A**	**HS** ***n* = 106**	**Controls** ***n* = 262**	***p***	**OR**	**95% CI**
01	19	56	0.57	0.82	0.45–1.42
02	54	132	0.99	1.01	0.70–1.46
03	28	69	0.92	1.00	0.63–1.61
11	8	43	0.05	0.44	0.20–0.95
23	9	19	0.85	1.18	0.52–2.65
24	23	40	0.21	1.47	0.86–2.53
25	8	12	0.38	1.67	0.67–4.15
26	6	16	0.94	0.92	0.36–2.40
**29**	**9**	**48**	**0.02**	**0.44**	**0.21–0.91**
**30**	**17**	**20**	**0.03**	**2.20**	**1.13–4.28**
31	6	13	0.99	1.14	0.43–3.06
32	11	12	0.07	2.33	1.01–5.38
33	5	16	0.79	0.77	0.28–2.12
34	0	1	N/A	N/A	N/A
66	2	4	1.00	1.37	0.29–6.50
68	5	22	0.28	0.59	0.23–1.52
80	2	1	0.20	4.14	0.54–31.58
Total *	212	524			
(**b**)					
**HLA-B**	**HS** ***n* = 106**	**Controls** ***n* = 262**	***p***	**OR**	**95% CI**
07	26	57	0.65	1.16	0.71–1.89
08	8	30	0.41	0.67	0.31–1.47
13	2	12	0.37	0.49	0.12–1.91
14	9	31	0.52	0.73	0.35–1.54
15	15	41	0.89	0.91	0.50–1.68
18	13	20	0.22	1.67	0.82–3.37
27	4	20	0.25	0.53	0.19–1.49
35	18	53	0.63	0.84	0.48–1.46
**37**	**6**	**4**	**0.04**	**3.64**	**1.08–12.24**
38	6	6	0.17	2.51	0.84–7.54
39	1	3	1.00	1.06	0.16–7.20
40	8	18	0.93	1.14	0.50–2.61
41	2	4	1.00	1.37	0.29–6.50
44	29	104	0.06	0.65	0.42–1.01
45	5	6	0.31	2.11	0.67–6.66
49	6	17	0.97	0.91	0.37–2.28
**50**	**9**	**10**	**0.04**	**0.38**	**0.15–0.92**
51	23	41	0.22	1.44	0.85–2.46
52	1	7	0.45	0.49	0.08–2.85
53	9	10	0.11	2.29	0.94–5.58
55	4	8	0.75	1.31	0.41–4.16
56	1	2	1.00	1.48	0.19–11.29
57	4	14	0.79	0.76	0.26–2.21
58	2	6	1.00	0.95	0.22–4.11
81	1	0	N/A	N/A	N/A
Total *	212	524			
(**c**)					
**HLA-C**	**HS** ***n* = 106**	**Controls** ***n* = 262**	***p***	**OR**	**95% CI**
01	8	18	0.99	1.10	0.47–2.58
02	9	36	0.24	0.60	0.28–1.27
03	16	48	0.58	0.81	0.45–1.46
04	43	75	0.06	1.52	1.01–2.31
05	17	37	0.77	1.15	0.63–2.09
06	17	36	0.70	1.18	0.65–2.15
07	51	134	0.73	0.92	0.64–1.34
08	9	28	0.66	0.79	0.36–1.69
12	11	26	0.95	1.05	0.51–2.16
14	6	6	0.19	2.51	0.80–7.89
15	11	26	0.95	1.05	0.51–2.16
16	11	50	0.07	0.52	0.26–1.02
17	2	4	1.00	1.24	0.23–6.81
18	1	0	N/A	N/A	N/A
Total *	212	524			

Note: *p*-values were not corrected for multiple comparisons. Antigens with statistically significant differences between the two groups are in bold. After the Bonferroni correction, significant *p*-values disappeared. * Note that total count is double the number of individuals due to the presence of two alleles.

**Table 2 jcm-09-03095-t002:** HLA-DRB1 (a), -DQA1 (b), and -DQB1 (c) allele frequencies in HS patients and healthy controls.

(**a**)					
**HLA-DRB1**	**HS** ***n* = 106**	**Controls** ***n* = 262**	***p***	**OR**	**95% CI**
01	25	68	0.75	0.90	0.55–1.46
03	13	52	0.13	0.59	0.32–1.11
04	33	62	0.21	1.37	0.87–2.17
**07**	**21**	**103**	**0.002**	**0.45**	**0.27–0.74**
08	10	19	0.63	1.31	0.60–2.88
09	1	5	0.68	0.49	0.06–4.24
10	3	6	0.72	1.24	0.31–5.00
11	31	49	0.05	1.66	1.03–2.69
12	2	6	1.00	0.95	0.22–4.11
13	38	74	0.24	1.33	0.87–2.04
14	6	10	0.62	1.50	0.54–4.17
15	26	62	0.97	1.04	0.64–1.70
16	3	8	1.00	0.93	0.24–3.52
Total	212	524			
(**b**)					
**HLA-DQA1**	**HS** ***n* = 106**	**Controls** ***n* = 262**	***p***	**OR**	**95% CI**
01	96	222	0.52	1.13	0.82–1.55
**02**	**21**	**102**	**0.0002**	**0.43**	**0.28–0.67**
03	33	75	0.75	1.10	0.71–1.72
04	8	15	0.68	1.33	0.56–3.19
05	52	109	0.31	1.24	0.85–1.80
06	2	1	0.20	4.98	0.45–55.22
Total	212	524			
(**c**)					
**HLA-DQB1**	**HS** ***n* = 106**	**Controls** ***n* = 262**	***p***	**OR**	**95% CI**
**02**	**29**	**141**	**0.0001**	**0.44**	**0.27–0.67**
**03**	**78**	**145**	**0.02**	**1.52**	**1.08–2.13**
04	9	16	0.56	1.41	0.61–3.24
05	42	95	0.67	1.12	0.74–1.67
06	54	127	0.80	1.07	0.74–1.54
Total	212	524			

Note: *p*-values were not corrected for multiple comparisons. After the Bonferroni correction, significant *p*-values remained significant except for DQB1*03 (DRB1*07 *p =* 0.026, DQA1*02 *p* = 0.0012, and DQB1*02 *p =* 0.0005).

**Table 3 jcm-09-03095-t003:** Distribution of DQB1*03:01 allele in HS patients and healthy controls and analysis of the DQB1*03:01 frequencies found in the different heterodimer molecules.

(**a**)						
**DQB1** ***03:01**		**HS *n* = 106**	**Controls *n* = 262**	***p***	**OR**	**95% CI**
		58	73	0.00001	2.33	1.57–3.44
TOTAL		212	524			
(**b**)						
**DQα**	**DQβ**	**HS *n* = 106**	**Controls *n* = 262**	***p***	**OR**	**95% CI**
03	03:01	15	14	0.01	2.77	1.31–5.85
05	03:01	41	58	0.004	1.92	1.24–2.98
06	03:01	2	1	0.20	4.80	0.45–55.22

In (**a**), the difference in the frequency of the DQB1*03:01 allele between the two groups reached a statistical power of 98%. In (**b**), only heterodimerization between DQα03/DQβ03:01 and DQα05/DQb03:01 showed statistical significance (*p* = 0.03 and 0.012 after the Bonferroni correction, respectively).

**Table 4 jcm-09-03095-t004:** Frequency of the HLA DRB1*07–DQBA1*02–DQB1*02 haplotype in HS patients and controls.

DRB1*07–DQA1*02–DQB1*02	HS *n* = 106		Controls *n* = 262	*p*	OR	95% CI
		16	89	0.0014	0.40	0.23–0.70
	Total	212	524			

After correction for multiple comparisons, *p*-value remained significant (*p* 0.028). The comparison included all the DRB1*07–DQA1*02–DQB1*02 haplotypes found in the current study.

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
