# Peer review of "Association of Human Leukocyte Antigens Class II Variants with Susceptibility to Hidradenitis Suppurativa in a Caucasian Spanish Population"

_jcm, 2020, doi:10.3390/jcm9103095_

Round 1

Reviewer 1 Report

The authors present an analysis of HLA association with hidradenitis suppurativa among 106 HS patients and 262 healthy controls from a Caucasian population of Northern Spain.  The main limitation to the study is the generalizability of the results. The study draws from a very limited patient population, which limits the generalizability of the results greatly.

The background information about hidradenitis at the beginning of the introduction can be made more concise since this is a dermatology journal and the reader can be expected to have familiarity with the disease.  Specifically, the first 3 paragraphs of the introduction can be condensed into a few sentences.

Introduction, grammatical: These statements should be separated: “forty-two unrelated Swedish patients with HS and 250 healthy controls these results found no association of any HLA alleles and genetic susceptibility to HS.” It should read: “forty-two unrelated Swedish patients with HS and 250 healthy controls. These results found no association of any HLA alleles and genetic susceptibility to HS.”

Introduction, grammatical: “The results obtained after analyze the data of the HLA alleles and haplotypes were used” should read “The results obtained after analysis of the data of the HLA alleles and haplotypes were used”

Section 2.1, grammatical: “we excluded patients with familial story of HS,” should read “we excluded patients with a familial history of HS,”

Section 2.1: Were other ethnicities beyond Caucasian excluded from the study or is your patient population exclusively Caucasian?

Author Response

We thank the Reviewer for his thoughtful comments that clearly improve the quality of this manuscript. Due to this, the manuscript has been modified accordingly. In addition, grammar corrections have been made.

 Point 1

- The authors present an analysis of HLA association with hidradenitis suppurativa among 106 HS patients and 262 healthy controls from a Caucasian population of Northern Spain.  The main limitation to the study is the generalizability of the results. The study draws from a very limited patient population, which limits the generalizability of the results greatly.

Response 1:

We thank the Reviewer for this comment that has been included at the end of the Discussion. Moreover, potential strengths of the study have also been incorporated to the Revised manuscript (lines 222-226) 

“The limited patient population may be a potential limitation of the study. However, our study has a number of strengths. In this regard, to our knowledge, it encompassed the largest series of HS patients ever evaluated for genetic susceptibility. Moreover, it had a monocentric design with the inclusion of patients homogeneously and carefully evaluated by a group of clinicians with high experience in the disease. “

Point 2

-The background information about hidradenitis at the beginning of the introduction can be made more concise since this is a dermatology journal and the reader can be expected to have familiarity with the disease.  Specifically, the first 3 paragraphs of the introduction can be condensed into a few sentences.

Response 2:

We agree with the Reviewer that HS is a very well-known disease among dermatologists. Consequently, we have shortened the first three paragraphs as suggested (Lines 39-48)

Point 3

-Introduction, grammatical: These statements should be separated: “forty-two unrelated Swedish patients with HS and 250 healthy controls these results found no association of any HLA alleles and genetic susceptibility to HS.” It should read: “forty-two unrelated Swedish patients with HS and 250 healthy controls. These results found no association of any HLA alleles and genetic susceptibility to HS.”

Response 3:

Again, we thank the Reviewer for helping us improve the grammar of our manuscript. We have separated these statements as suggested: “forty-two unrelated Swedish patients with HS and 250 healthy controls. These results found no association of any HLA alleles and genetic susceptibility to HS.” (line 66)

Point 4

-Introduction, grammatical: “The results obtained after analyze the data of the HLA alleles and haplotypes were used” should read “The results obtained after analysis of the data of the HLA alleles and haplotypes were used”

Response 4:

The sentence has been reformulated following the suggestion of the Reviewer: “The results obtained after analysis of the data of the HLA alleles and haplotypes were used”. (line 73)

Point 5

-Section 2.1, grammatical: “we excluded patients with familial story of HS,” should read “we excluded patients with a familial history of HS,”

Response 5:

Again, we have changed the sentence as suggested by the Reviewer: “we excluded patients with a familial history of HS (line 80)

Point 6

-Section 2.1: Were other ethnicities beyond Caucasian excluded from the study or is your patient population exclusively Caucasian?

Response 6:

We thank the Reviewer for this question that we are pleased to address in the Revised version of the manuscript: “No other ethnicities were included in our patient population” (lines 83-84)

Reviewer 2 Report

This is an intersting paper with some nobvel results. Theu extent the current knowledge in the field. I enjoyed reading the manuscript.

Author Response

Comments and Suggestions for Authors

This is an interesting paper with some novel results. They extent the current knowledge in the field. I enjoyed reading the manuscript.

Response

We greatly appreciate the comments raised by the Reviewer on our work.

Reviewer 3 Report

The article is well written , interesting and highlights a possible association between HLA class II and HS.

My major concern is  the statistical part:

You said that you used the Bonferroni correction in order to approach  the multiple comparison problem. I was however not able to find the p value that you considered significant. Since this article is basicly based on multiple comparison, I think that you should clearly calculate and  indicate the p value that you consider significant within the text.

Author Response

Comments and Suggestions for Authors

The article is well written, interesting and highlights a possible association between HLA class II and HS.

Point 1

My major concern is  the statistical part:

You said that you used the Bonferroni correction in order to approach  the multiple comparison problem. I was however not able to find the p value that you considered significant. Since this article is basicly based on multiple comparison, I think that you should clearly calculate and  indicate the p value that you consider significant within the text.

Response 1

We also thank the Reviewer 3 for his/her thoughtful comments on our study.

We are pleased to address his/her potential point of concern.

The Bonferroni correction was calculated by multiplying the initial p-value by the number of HLA alleles at each locus. A value of p <0.05 was considered statistically significant.

Consequently, in the Revised manuscript we have added the following sentence:

“It was calculated by multiplying the initial p-value by the number of HLA alleles at each locus. A value of p <0.05 was considered statistically significant.“ (lines 104-106)

Round 2

Reviewer 1 Report

Please clarify if the healthy control blood donors were examined by a dermatologist as well.  HS often goes undiagnosed for years.  To ensure no control patients had undiagnosed HS, it would be important for a dermatologist to examine their intertriginous skin.  Please clarify whether or not this was performed.

Author Response

2. Experimental Section

2.1. Subjects

Comments and Suggestions for Authors

Please clarify if the healthy control blood donors were examined by a dermatologist as well.  HS often goes undiagnosed for years.  To ensure no control patients had undiagnosed HS, it would be important for a dermatologist to examine their intertriginous skin.  Please clarify whether or not this was performed.

Response.

We thank the Reviewer for this comment that we should have included before. Due to this, the manuscript has been modified accordingly (Lines 84-85)

Healthy control blood donors were examined by a dermatologist to rule out the presence of skin lesions that could bias the study